# Targeting Canonical and Non-Canonical STAT Signaling Pathways in Renal Diseases

**DOI:** 10.3390/cells10071610

**Published:** 2021-06-27

**Authors:** Lili Gai, Yuting Zhu, Chun Zhang, Xianfang Meng

**Affiliations:** 1Department of Nephrology, Union Hospital, Tongji Medical College, Huazhong University of Science and Technology, Wuhan 430022, China; m201975632@hust.edu.cn (L.G.); drzhuyuting@hust.edu.cn (Y.Z.); 2Department of Neurobiology, School of Basic Medical Sciences, Tongji Medical College, Huazhong University of Science and Technology, Wuhan 430030, China

**Keywords:** renal diseases, STAT, signal transduction

## Abstract

Signal transducer and activator of transcription (STAT) plays an essential role in the inflammatory reaction and immune response of numerous renal diseases. STATs can transmit the signals of cytokines, chemokines, and growth factors from the cell membrane to the nucleus. In the canonical STAT signaling pathways, upon binding with their cognate receptors, cytokines lead to a caspase of Janus kinases (JAKs) and STATs tyrosine phosphorylation and activation. Besides receptor-associated tyrosine kinases JAKs, receptors with intrinsic tyrosine kinase activities, G-protein coupled receptors, and non-receptor tyrosine kinases can also activate STATs through tyrosine phosphorylation or, alternatively, other post-translational modifications. Activated STATs translocate into the nucleus and mediate the transcription of specific genes, thus mediating the progression of various renal diseases. Non-canonical STAT pathways consist of preassembled receptor complexes, preformed STAT dimers, unphosphorylated STATs (U-STATs), and non-canonical functions including mitochondria modulation, microtubule regulation and heterochromatin stabilization. Most studies targeting STAT signaling pathways have focused on canonical pathways, but research extending into non-canonical STAT pathways would provide novel strategies for treating renal diseases. In this review, we will introduce both canonical and non-canonical STAT pathways and their roles in a variety of renal diseases.

## 1. Introduction

Signal transducer and activator of transcription (STAT) is a family of latent transcription factors involved in the signaling of diverse ligands such as cytokines, chemokines, and growth factors. The signals can be transmitted from the cell membrane to the nucleus by initiating caspase of tyrosine kinases and STATs phosphorylation and activation, binding to response elements of DNA and inducing the transcription of target genes [1]. Thus, the STAT signaling pathways enable cells to rapidly initiate a transcriptional response to external stimulation. Given the mechanisms of STATs, they are involved in many inflammatory and immune related diseases including cancers and some autoimmune diseases such as systemic lupus erythematosus, inflammatory bowel diseases, and rheumatological diseases [2,3,4,5].

The STAT family includes seven mammalian STATs: STAT1, STAT2, STAT3, STAT4, STAT5a, STAT5b, and STAT6, which share several similar domains: N-terminal homotypic dimerization domains involved in dimerization/tetramerization; coiled-coil domains involved in interaction with other proteins; DNA binding domain (DBD); linker domain; SRC homology 2 (SH2) domain; phosphor–tyrosine residue; transactivation domain [6]. STATs can be activated by different ligands and mediate different biological events, which are shown in Table 1 [1,7].

Current studies have mostly focused on the canonical Janus kinase (JAK)/STAT pathways. However, some studies have shown that non-canonical pathways also play an important role in the development of renal diseases. Therefore, in this review we will illustrate both the canonical and non-canonical STAT pathways in order to provide new insights to develop effective interventions for renal diseases.

## 2. The Canonical STAT Signaling Pathways

### 2.1. The Mechanisms of Canonical STAT Signaling Pathways

STATs mainly transmit the signals of cytokines in canonical STAT pathways. However, cytokine receptors do not have the capacity of intrinsic tyrosine kinases. Instead, the receptor-associated tyrosine kinases, mostly referred to as JAKs, are activated when ligands bind to cytokines [13]. The JAK family consists of four members: JAK1, JAK2, JAK3, and tyrosine kinase 2 (TYK2). JAKs selectively bind to the cytoplasmic domains of various cytokine receptors and participate in various biological events [14]. The cytokines bind to cognate receptors and cause the receptors dimerization or oligomerization. Then, the receptor-associated tyrosine kinases JAKs transphosphorlate and activate each other. The activated JAKs phosphorylate the tyrosine residues in the intracellular segment of the receptors to expose the docking sites, STATs are recruited to the receptor complex through their SH2 domains, then the activated JAKs phosphorylate the conserved tyrosine of STATs, mostly Tyr705. Phosphorylated-STATs (pSTATs) are dissociated from the receptor and form homodimerization or heterodimerization through the binding of the phosphorylated tyrosine to the SH2 domains of one monomer or other STATs. Then, STAT dimers transfer to the nucleus, bind to specific DNA elements, and activate the transcription of cytokine-responsive genes. Activated phosphorylated STATs exist for a short time until dephosphorylated by the protein tyrosine phosphatases (PTPs) in the nucleus, sending the signal to export the STATs into the cytoplasm [1,7] (Figure 1).

STATs can also be activated via several alternative pathways: (i) receptors with intrinsic tyrosine kinase activities such as EGFR (epidermal growth factor receptor), PDGFR (platelet-derived growth factor receptor), and FGFR (fibroblast growth factor receptor), which can activate STATs dependently or independently by JAKs. Without EGF (epidermal growth factor) stimulation, JAK- and Src-mediated constitutive activation of STAT3; upon EGF stimulation, EGFR kinases induced maximal activation of STAT signaling in breast cancer cells [15]. (ii) G-protein coupled receptors, including chemokine receptors, can activate STATs after binding to their cognate ligands [16]. (iii) Non-receptor tyrosine kinases such as oncoproteins Src family kinases (SFK) can induce constitutive STAT activation [1]. SFK is a family of proto-oncogenic non-receptor tyrosine kinases consisting of members: c-Src (Sarcoma), Blk (B-lymphoid tyrosine kinase), Fgr (Gardner-Rasheed feline sarcoma), Fyn, Frk (Fyn-related kinase), Hck (Hematopoietic cell kinase), Lck (Lymphocyte specific kinase), Lyn, Yes (Yamagichi sarcoma), and Yrk (Yes-related kinase) [17]. Of all the SFKs, the function of Src has been well studied. Src can interact with protein-tyrosine kinase receptors, integrin receptors, G-protein coupled receptors, antigen receptors and Fc receptors, cytokine receptors, and steroid hormone receptors at the plasma membrane. Phosphorylation by an adjacent Src molecule at Tyr419 in SH1 domain results in activation of Src, whereas phosphorylation by C-terminal Src kinase (Csk) or Chk (Csk homologous kinase) at Tyr530 in COOH-terminal region results in its inactivation [18]. Activated Src can directly induce phosphorylation and activation of STATs.

### 2.2. The Alternative STAT Post-Translational Modifications

In addition to typical tyrosine phosphorylation, STATs can also be regulated by other post-translational modifications: serine phosphorylation, acetylation, methylation, ubiquitination, SUMOylation, and glycosylation. Serine phosphorylation of STATs can induce maximal transcriptional activity and influence tyrosine phosphorylation [19]. Acetylation of STATs is dependent on the balance between histone deacetylases (HDACs) and histone acetyltransferases (HATs), and functions as a positive or negative regulator through increasing DNA binding affinity and transcriptional activation, promoting protein–protein interaction or modulating dimerization [20]. The lysine methylation of promoter-bound STAT3 decreased the amount of steady-state activated STAT3 and the expression of some STAT3-activated genes [21]. Linear ubiquitination of STAT1 could inhibit STAT1 binding to the type-I IFNR and restrict STAT1 activation [22]. STAT1 SUMOylation on Lys703 could inhibit STAT1 transcriptional activity by disrupting STAT1 tyrosine phosphorylation and dimerization [23], while STAT3 SUMOylation on Lys451 could inhibit STAT3 transcriptional activity by recruiting a nuclear phosphatase known to dephosphorylate STAT3 [24]. The mutated STAT5 without O-linked-N-acetylglucosaminylation (a type of glycostlation) resulted in decreased tyrosine phosphorylation, oligomerization, and transactivation capacity [25]. The crosstalk among diverse post-translation modifications of STATs may guide researchers to concentrate on the mechanisms of STATs.

## 3. The Non-Canonical STAT Signaling Pathways

### The Mechanisms of Non-Canonical STAT Signaling Pathways

Non-canonical STAT signaling pathways consist of preassembled receptor complexes, preformed STAT dimers, unphosphorylated STATs (U-STATs), and non-canonical functions including mitochondria modulation, microtubule regulation, and heterochromatin stabilization [26] (Figure 2).

Cytokine receptors might dimerize and form preassembled receptor complexes in the absence of ligands, which was verified by the observation that ectodomains of unliganded erythropoietin receptor were crystallized as a homodimer [27]. Similarly, STATs might dimerize and form preformed STAT dimers in the absence of the activating tyrosine, which was based on a series of techniques including size exclusion chromatography, native gel electrophoresis, cross-linking, and both velocity and equilibrium sedimentation studies [28]. U-STAT proteins are constantly shuttling between cytoplasmic and nuclear compartments. The cytoplasmic U-STATs are upregulated due to the activation of the STATs in response to ligands, and can function as transcription factors in the absence of tyrosine phosphorylation [29]. The latent nuclear U-STATs molecules consistently located in the nucleus can directly contribute to gene regulation [30].

STATs can exert non-canonical roles without tyrosine phosphorylation and DNA-binding [30]. (i) Mitochondria modulation: Gough et al. discovered that mitochondrial STAT3 (mitoSTAT3) supported Ras-dependent oncogenic transformation by augmenting electron transport chain (ETC) activity [31]. Studies indicated that STAT3 entered into mitochondria through GRIM-19 (a component of Complex I of the ETC) [32] and the mitochondrial importer Tom20 [33]. Xu et al. demonstrated that mitochondrial translocation of STAT3 was also promoted by its acetylation [34]. The function of mitoSTAT3 is dependent on C-terminal serine phosphorylation but independent of its tyrosine phosphorylation, DNA-binding domain, SH2 domain, and nuclear translocation. Although both p-Tyr-STAT3 and p-Ser-STAT3 are both present in mitochondria, serine phosphorylation seems to be more critical for regulation of ETC activity by mitoSTAT3. P-Ser-STAT3 in mitochondria plays its role through associating with Complex I of the ETC and promoting Complex I activity and mitochondrial respiration, associating with Complex II or Complex V (ATP synthase) [31], promoting the production of ATP, increasing mitochondrial Ca2+ [35], inhibiting the opening of the mitochondrial permeability transition pore (MPTP) [33], and reducing mitochondrial reactive oxygen species (ROS) production [36]. In addition to its role in cancer biology, mitoSTAT3 could protect the heart against ischemia/reperfusion injury [33], promote nerve growth factor induced neurite outgrowth [37], and increase axon regrowth [38]. MitoSTAT3 was upregulated following spinal cord injury [39]. CD4 cells activated with IL-6 increased mitochondrial Ca^2+^ via mitoStat3 and contributes to a sustained expression of cytokines [35]. MitoSTAT3 participated in memory generation of CD8 (+) T cell and antibody production of B cell mediated by IL-21 [40]. (ii) Microtubule regulation: STAT3 modulates the microtubule dynamics and cell migration by binding to the COOH-terminal tubulin-interacting domain of stathmin and antagonizing its microtubule destabilization activity in the cytoplasm [41]. (iii) Heterochromatin stabilization: Nuclear U-STATs are located on heterochromatin in association with heterochromatin protein 1 (HP1), maintaining heterochromatin stability. Activation and phosphorylation of STAT can cause STAT dispersal from heterochromatin, leading to HP1 displacement and heterochromatin destabilization.

The most studied non-canonical aspect of STAT signaling pathways is mitochondria STAT3. Studies have revealed that mitoSTAT3 plays an essential role in neuron cells and ischemia/reperfusion cardio injury [33,37]. Therefore, the role of mitoSTAT3 in podocytes and kidney ischemia/reperfusion injury remains to be further explored. In view of the function of mitoSTAT3 in immune response, mitoSTAT3 could be a target for chronic inflammatory diseases including renal diseases. Although studies on the non-canonical JAK/STAT pathways are limited and controversial, they still enrich our current knowledge of the roles of STAT pathways in renal diseases from diverse aspects.

## 4. The STAT Signaling Pathways and Renal Diseases

### 4.1. Acute Kidney Injury

Inflammation, microvascular dysfunction, and renal tubular injury influence and deteriorate each other, forming a vicious cycle that culminates in acute kidney injury (AKI) exacerbation [42]. STAT signaling pathways are also involved in AKI. However, the role of STATs in AKI varies in different animal models and cell types. Endothelial STAT3 played a protective role in the ischemia-reperfusion (I/R) model of AKI [43]. In contrast, cisplatin could lead to AKI deterioration by activating the JAK2/STAT3 pathway [44]. Anyway, targeting STAT signaling pathways can be a promising therapy for treating AKI.

Cytokines and their receptors are essential for JAK/STAT pathways and, therefore, are potential targets for AKI treatment. IL-6 is a common cytokine activating JAK2/STAT3 pathway. PTX3 (The long pentraxin 3) could suppress I/R-induced interstitial fibrosis and aspirin-triggered resolvin D1 could protect against LPS-induced inflammation in AKI through decreasing IL-6 level and blocking the IL-6-STAT3 axis [45,46]. IL-19 contributed to AKI by activating STAT3 [47]. In contrast, IL-4/IL-13 and IL-22 attenuated AKI by activating JAK3/STAT6 and JAK2/STAT3 pathways, respectively [48,49]. Chemokine (C-X-C motif) ligand 8 (CXCL8/IL-8) could bind to G-protein coupled receptors (CXCR1 and CXCR2). G31P, an CXCR1 and CXCR2 antagonist, inhibited the phosphorylation of JAK2/STAT3, leading to alleviation of sepsis-induced AKI [50]. Gefitinib (a selective inhibitor of EGFR tyrosine kinase) ameliorated renal fibrosis and macrophage infiltration by inhibiting the EGFR-STAT3-HIPK2 axis in AKI [51]. Directly targeting kinases and STATs can also influence the progression of AKI. Using AG490 (a JAK2 inhibitor) could alleviate cisplatin-induced AKI through suppression of the JAK2/STAT3 pathway [44]. On the contrary, using a relatively selective JAK3 inhibitor, tofacitinib, led to more severe AKI by delaying recovery from AKI [48]. I/R mice models with endothelial STAT3 depletion significantly exacerbated kidney dysfunction in AKI [43]. Edaravone could mitigate the loss of mitochondrial membrane potential and mitochondrial microstructural damage in I/R-induced kidney injury by inhibiting the JAK/STAT pathway [52].

STAT signaling pathways play different roles in the pathological process of AKI, which are briefly summarized in Table 2. In aggregate, STAT signaling pathways represent an attractive therapeutic target to treat AKI, and more research focused on non-canonical STAT pathways is desirable.

### 4.2. Focal Segmental Glomerulosclerosis

Focal segmental glomerulosclerosis (FSGS) is a progressive glomerulopathy that is one of the primary causes of end-stage renal diseases (ESRD) [53]. HIV associated nephropathy (HIVAN) is a type of FSGS, which is characterized by collapsing focal segmental glomerulosclerosis. Clinical investigation has elucidated that JAK/STAT pathways are activated in peripheral blood mononuclear cells (PBMCs), glomeruli, and renal tubulointerstitium of patients with FSGS, particularly STAT1 and STAT3 pathways [54]. Liang et al. found that JAK2/STAT3 pathways stimulated by cytokines such as TGFβ1, connective tissue growth factor (CTGF), and fibronectin (FN) might be responsible for abnormal accumulation of extracellular matrix (ECM) and myofibroblast transdifferentiation in FSGS [55]. FSGS serum and cardiotrophin-like cytokine factor 1 (CLCF1) increased albumin permeability (P_alb_) and altered actin cytoskeleton of podocytes through the JAK2/STAT3 pathway, which could be blocked by anti-CLCF1 monoclonal antibody, heterodimer CLCF1-CRLF1(cytokine receptor-like factor-1, a cytokine dimerized with CLCF-1), JAK2 inhibitor BMS-91154, and STAT3 inhibitor Stattic [56,57]. HIV-1 accessory protein Nef promoted podocyte dedifferentiation and proliferation through activating Src-STAT3 in HIVAN. A dominant negative mutant of Src abolished the Nef’s effects, while reduced STAT3 activity with a homozygous serine to alanine mutation in Ser727 (Tg26-SA/-) and incomplete deletion of podocyte STAT3 could both reduce Nef’s effects [58,59,60]. Suppression of sirtuin 1 (SIRT1) deacetylase led to enhanced acetylation and activation of STAT3 in HIVAN kidneys. Using SIRT1 agonist BF175 decreased acetylation of STAT3, resulting in kidney injury attenuation in HIVAN [61].

These findings illustrate the relationship between FSGS and STAT signaling pathways, and provide clinical and basic mechanistic insights to treat FSGS through targeting STAT signaling pathways.

### 4.3. IgA Nephropathy

IgA nephropathy (IgAN) is also a progressive glomerulopathy. It is characterized by IgA-dominant or co-dominant immune deposits in the mesangial area of the glomeruli [62]. Arakawa et al. found that JAK/STAT pathways participated in the development of IgAN [63]. In kidneys of IgAN patients, immunohistochemistry staining for JAK-2 and pSTAT1 in glomerular mesangium and endothelium, JAK2, pSTAT1, and pSTAT3 in the tubulointerstitial were increased [64]. Yamada et al. found that IL-6/JAK2/STAT3 mediated overproduction of galactose-deficient IgA1 (the key pathogenic molecule in IgAN) in IgA1-secreting cells from patients with IgAN, which could be reduced by STAT3 inhibitor stattic and JAK2 inhibitor AZD1480 [65]. In addition, IL-4/STAT6 mediated aberrant glycosylation of IgA1 in tonsillar mononuclear cells of IgAN patients [66]. Studies investigating STAT pathways on IgAN are mostly limited to immune cells, which are related to the pathological mechanisms of IgAN and lack of typical animal models. Thus, further studies are needed to investigate this application in renal cells and in vivo.

### 4.4. Lupus Nephritis

Systemic lupus erythematosus (SLE) is a typical autoimmune disease affecting multiple organs, 50% of which can affect the kidney, called lupus nephritis (LN) [67]. LN is a primary risk factor for the morbidity and mortality of SLE, 30% of which will develop into ESRD, even if treated with current anti-inflammatory and immunosuppressive therapies [68]. As typical pathways involved in inflammatory reaction and immune response, JAK/STAT pathways can increase levels of anti-dsDNA antibodies, complement C3 and IgG deposition in glomeruli, frequencies of autoantigen-specific Ab-secreting cells, T-cell and macrophage infiltration, and can decrease levels of serum complement C3 after being activated by cognate ligands. AG490 and CEP-33779 (both selective JAK2 inhibitors) and CP-690,550 (a JAK3 inhibitor) could mitigate the development and progression of LN [69,70,71]. Clinical trials are exploring the relationship between JAK3 levels in the serum and histopathological classes of LN in lupus patients during the active disease stage (ClinicalTrials.gov, Identifier: NCT04293510). Stattic and S3I-201, both STAT3 inhibitors, could achieve a therapeutic effect by delaying the onset of LN. Interestingly, pSTAT1 further elevated after administration of S3I-201. It was speculated that STAT3 was more involved in LN than STAT1, and STAT1 might play a protective role in LN [72,73]. Moreover, Yoshida et al. found that T-cell-specific silencing of STAT3 was a more optimal therapy than nonspecific STAT3 silencing in inhibiting B cell activation, autoantibody production, and kidney cellular infiltration in LN [74]. Besides, STAT6 deficiency or anti-IL-4 antibody inhibited the progression of LN, while STAT4 deficiency contributed to the development of LN [75]. Some biological extracts or newly synthesized substances might alleviate LN by targeting JAK/STAT pathways. For example, artesunate (an anti-malarial drug) ameliorated the LN symptoms by inhibiting JAK2/STAT3 [76]. Synthetic triterpenoid CDDO-Me suppressed the onset and development of LN, partly by inhibiting JAK1/STAT3 [77].

These studies demonstrate that drug targeting JAK/STAT pathways have potential value of application in LN therapy. Although there are clinical trials associated with JAK/STAT pathways in SLE and LN, no drug has been approved to use in clinical treatment. These potential treatments still need to be further investigated.

### 4.5. Diabetic Nephropathy

Diabetic nephropathy (DN) is one of the most serious vascular complications of diabetes mellitus that can develop into ESRD. It is characterized by glomerular mesangial expansion, podocyte loss, tubulointerstitial injury, and fibrosis [78]. DN still lacks effective treatments. Microarray analysis of renal tissues from subjects with DN has demonstrated that the expression of JAK-1,2,3 and STAT-1,3 are increased in glomeruli and tubular area [79]. Under a high glucose (HG) environment, cytokines, overactivated RAS system, ROS, and advanced glycation end products (AGE) are main factors that can influence JAK/STAT signaling pathways. The role of JAK/STAT pathways in DN varies with different cytokines and the duration of cytokine stimulation, different STATs, and downstream signaling pathways [80]. IFN-γ and IL-17A can play protective roles. IFN-γ could suppress renal interstitial fibrosis and attenuate mesangial matrix accumulation through activating STAT1 [81,82]. IL-17A decreased in patients with advanced DN. Low-dose IL-17A administration could suppress renal fibrosis and inflammation through inhibiting STAT3 [83]. In contrast, IL-6, OSM, and IL-1β function as pro-inflammatory and pro-fibrotic molecules to promote DN progression by activating STA1/3 pathways. IL-6 could induce mesangial expansion, glomerular cell proliferation, podocyte hypertrophy, and renal inflammation [84]. OSM and IL-1β promoted tubular epithelial cell-myofibroblast transdifferentiation (TEMT) [85,86]. Moreover, IL-15 plus lipopolysaccharide (LPS) promoted inflammation by downregulating vitamin D receptor (VDR) expression and upregulating p-STAT5 expression in monocytes incubated with sera from DN patients. The pro-inflammatory effects of IL-15 and LPS could be blocked by 1,25-(OH)_2_D_3_ [87]. Angiotensin II (Ang II) could induce JAK2, STAT1, STAT3, and STAT5 phosphorylation, which could be enhanced by HG stimulation, promoting GMC cell growth and collagen IV synthesis. [16] HG and Ang II can also induce a rapid increase in intracellular ROS via activating the polyol pathway and can stimulate IL-6 secretion. Overexpressed ROS and IL-6 can promote HG and Ang II-induced JAK2-STAT1/3 pathway phosphorylation and activation, then upregulate mesangial expansion, extracellular matrix (ECM) production, and collagen synthesis [88,89]. The formation of AGE is accelerated under HG-induced oxidant stress conditions. AGE could induce podocyte apoptosis and inflammation through CXCL9-mediated JAK2/STAT3 phosphorylation [90]. AGE could also induce STAT3 acetylation in human podocytes [91]. The activation of JAK/STAT pathways and the progression of DN could be alleviated by Ang II receptor antagonist valsartan, the HMG-CoA reductase inhibitor simvastatin, siRNA-CXCL9, STAT1 or STAT3 ‘decoy’ oligodeoxynucleotides (prevented STAT1 and STAT3 from binding to DNA), JAK2-specific inhibitor AG-490, SIRT1 agonist, and small-molecule BET-specific bromodomain inhibitor (MS417) [90,91,92,93]. In addition, reduced STAT3 activity in Tg26-SA/- mice could prevent diabetic glomerulopathy [94]. Under diabetic conditions, excessive DPP4 would facilitate the transformation of its substrate SDF-1α into cleaved form, thus suppressing the SDF-1α/CXCR4/STAT3 pathway and impeding the translocation of STAT3^S727^ into mitochondria. Diminished interaction of STAT3 and the mitochondrial fusion-associated protein OPA1 contributed to mitochondrial dysfunction in diabetic tubular cells, inducing DN progression [95].

Among all renal diseases, the role of STAT signaling pathways in DN is the most widely studied. As downstream pathways of numerous signal molecules, STAT pathways play a vital role in the progression and exacerbation of DN; therefore, the inhibitors of STAT pathways may show remarkable effects in DN treatment. Besides traditional treatments, treatments related with non-canonical STAT signaling pathways need to be explored.

### 4.6. Renal Fibrosis

Renal fibrosis is one of the main characteristics of chronic renal diseases. Unilateral ureteric obstruction (UUO) is a common obstructive nephropathy model of renal fibrosis [96]. In vivo, IL-6, LIF (leukemia inhibitory factor) and other ligands promoted fibrotic markers such as type I collagen, FN, and a-smooth muscle actin (α-SMA) overexpression, promoting extracellular matrix overproduction and interstitial myofibroblasts proliferation by inducing JAK/STAT3 activation in UUO models [97,98]. Besides JAKs, non-receptor tyrosine kinases Fyn and Src were upregulated in UUO models. Inhibition of Fyn attenuated renal fibrosis through inhibiting p-STAT3 [99]. Dasatinib and nifuroxazide mitigated UUO induced-renal fibrosis through inhibiting Src/STAT-3/NF-κB signaling [100,101]. In addition, the role of STAT3 in UUO was further verified by S3I-201, a STAT3 inhibitor which inhibited STAT3 phosphorylation and attenuated renal fibrosis [98]. As well, STX-0119, a new inhibitor of STAT3 dimerization, inhibited renal fibrotic genes expression without affecting STAT3 phosphorylation [102]. Resveratrol (SIRT1 activator) decreased STAT3 acetylation on Lys685, resulting in inhibition of Ang II-induced upregulation of STAT3 Try705 phosphorylation and downregulation of pro-fibrotic genes in UUO models [103]. However, in SOCS3(+/−) UUO mice, STAT3 was increased with markedly suppressed renal fibrosis that was aggravated by pre-treatment with JAK inhibitor-incorporated nanoparticles (pyridine6-PGLA). It is concluded that JAK/STAT3 signaling may promote the repair process of renal fibrosis in UUO [104]. Besides STAT3, Yukawa et al. found that STAT6 was also involved in renal fibrosis. STAT6 exerted a protective role on renal cell apoptosis but promoted renal fibrosis by activating collagen synthesis following UUO [105]. Zhou et al. also found that H2S could inactivate IL-4/STAT6 pathways by suppressing NLRP3 signaling and thus alleviate renal fibrosis in UUO [106].

Based on these findings, STATs may play a dual role in renal fibrosis, but most studies predispose to its profibrotic effects. Targeting STAT pathways may be a promising therapy for renal fibrosis.

### 4.7. Autosomal Dominant Polycystic Kidney Disease

Autosomal dominant polycystic kidney disease (ADPKD) is an inherited renal disease with multiple excessive proliferation of epithelial-lined cysts in both kidneys, which can deteriorate renal function progressively. The major cause is PKD1 gene mutation, which encodes polycystin-1 (PC1) [107]. During renal development, membrane-anchored full-length PC1 might cause direct activation of STAT1 and STAT3 via JAK2. While, during the progression of ADPKD, the cleaved PC1 tail could co-activate STAT1 that had been activated by IFN, STAT3 that had been activated by IL-6, EGF, HGF (hepatocyte growth factor) signaling, STAT6 that had been activated by IL4, IL-13 signaling to promote renal cyst growth and fibrosis [108]. Src inhibitor SKI-606 could reduce renal cyst growth [108]. Constitutive activation of PC1-p30 (a proteolytic fragment of PC1 corresponding to the cytoplasmic tail)/Src/STAT3 pathways would be amplified after integrating signaling inputs from EGF and cAMP, and lead to aberrant proliferation of renal epithelial cells and cyst growth in ADPKD [109]. Genetic knockout of STAT3 and STAT6, STAT3 inhibitor S3I-201, STAT6 non-specific inhibitor teriflunomidelysine, methyltransferase SMYD2 specific inhibitor AZ505, or anti-parasitic compound pyrimethamine could attenuate ADPKD cell proliferation and reduce renal cyst formation [108,110,111]. Fragiadaki et al. found that loss of STAT5 could reduce renal cyst growth and overexpression of growth hormone (GH) would activate STAT5 in ADPKD. Therefore, the GH/STAT5 axis might be a novel therapeutic target in ADPKD [112]. The relationship of STATs and ADPKD has aspects that can be further explored.

### 4.8. Renal Cancers

The persistent cytokine and growth factor signaling could lead to constitutive STAT activation though JAK1, JAK2, and Src in several types of cancer, including renal cancers [113,114,115,116]. Blocking constitutive STAT signaling could make cells more susceptible to apoptosis and suppress tumor growth by suppressing expression of several antiapoptotic proteins such as BCL2, BCL-XL, MCL1, and survivin [117]. Renal cell carcinoma (RCC) constitutes more than 90% of primary renal cancers and is a typical immunogenic tumor [118]. Thus, we mainly illustrate the roles of STAT signaling pathways in RCC.

Treatment with IL-6R antibody tocilizumab in IFN-α-stimulated RCC cells enhanced pSTAT1 expression and inhibited SOCS3 and pSTAT3 expression. Combination therapy using tocilizumab with IFN could suppress tumor growth [119]. MicroRNA-363 could inhibit the JAK2-STAT3 axis though specifically binding to GHR and downregulating the expression of GHR, leading to the inhibition of the angiogenesis, proliferation, invasion, and migration in RCC [120]. Dihydrotestosterone promoted kidney cancer cell proliferation by activating the STAT5 pathway via androgen and glucocorticoid receptors (AR and GR), which could be attenuated by AR and GR knockdown [115]. Non-coding RNA 886 also promoted RCC growth and metastasis by activating the JAK2/STAT3 pathway, which was attenuated by JAK2 inhibitor AG490 [121]. Combined Src-Stat3 inhibition using Src inhibitors dasatinib and JAK/STAT inhibitors CYT387 synergistically reduced cell proliferation and increased apoptosis in RCC cells [122]. Inhibiting STAT1 signaling by using fludarabine or STAT1 siRNA would increase radiosensitization in RCC cell lines [123]. Using STAT3 inhibitors or STAT3 small interfering, RNA could also inhibit RCC cells proliferation, migration, and invasion [124]. STAT6 mediated IL-4-induced growth inhibition in RCC cells, which was abrogated by si-STAT6 [125]. SIRT1 could repress the STAT3-FGB axis to inhibit RCC tumorigenesis by deacetylating STAT3, which led to STAT3 destabilization and degradation [126].

Both canonical and non-canonical STAT signaling pathways are extensively studied and play an important role in renal cancers. According to the research, combined use of different STAT signaling inhibitors may be a more viable strategy in the treatment of renal cancers.

### 4.9. Other Renal Diseases

Mesangial proliferative glomerulonephritis (MsGN) is a common progressive glomerular disease and is characterized by glomerular mesangial cell (GMC) proliferation. SOCS1 was decreased in the kidneys of MsGN animal models with increased STAT1 expression. Overexpression of SOCS1 or inhibition of STAT1 could suppress IFN-γ-induced MHC class II expression in mesangial cells and ameliorate renal injury in MsGN [127]. IL-10 could activate MC STAT3 and induce MC proliferation. Using tellurium compound AS101 could ameliorate MC proliferation by inhibiting IL-10. [128]

Successful kidney transplantation brings a better quality of life. However, brain death (BD)-induced renal injury and postoperative immunosuppression bring great challenges to kidney transplantation. The immunosuppressive effect of JAK1 and JAK3 inhibitor tofacitinib in renal transplantation has been confirmed in clinical trials; however, the serious side effects led to the discontinuation of its development for transplantation [129]. Li et al. found that ginkgo biloba extract EGb761 could protect donor kidneys from BD-induced damages by partly blocking JAK/STAT3 signaling pathways [130]. Therefore, taking JAK/STAT as the research direction may enlighten more immunosuppressive treatment.

Coronavirus disease 2019 (COVID-19), a newly emerged and rapidly expanding pandemic caused by severe acute respiratory syndrome coronavirus-2 (SARS-CoV-2 virus), is characterized by acute tubular necrosis and interstitial inflammation. Kidney complications might occur owing to cytokine release syndrome (CRS), secondary injury from acute respiratory distress syndrome (ARDS), or the use of renal-toxic therapies, which are associated with the severity of COVID-19 and mortality rates. There is no specific treatment for COVID-19–associated AKI currently [131]. Matsuyama et al. recently found that SARS-CoV-2 gene products, the NSP1 and ORF6 proteins, could induce STAT1 dysfunction and compensatory hyperactivation of STAT3 [132]. Thus, JAK/STAT inhibition may present an attractive therapeutic strategy for COVID-19-associated AKI. 

To sum up, most studies on renal diseases have focused on canonical STAT pathways. Research on non-canonical STAT pathways in renal diseases is still limited. Turning attention to these non-canonical pathways or combining them with canonical pathways may bring new ideas to the treatment of renal diseases.

## 5. Endogenous Inhibitors of STAT Signaling Pathways

STAT signaling pathways can be inactivated by negative endogenous regulators. PTPs, suppressors of cytokine signaling (SOCS), and protein inhibitors of activated STAT (PIAS) can suppress STAT pathways via different mechanisms (Figure 1). Targeting STAT endogenous inhibitors may be a potential therapy for renal diseases.

PTPs dephosphorylate and dissociate phosphorylated STAT dimers in the nucleus and export monomers into the cytoplasm, controlling the magnitude and duration of STAT signaling. Studies on PTPs have mainly concentrated on tumors. Wiede et al. found that PTPN2 deletion in T-cells promoted anti-tumor immunity and CAR T-cell efficacy in solid tumors by activating Src family kinase Lck and STAT5 signaling [133]. Song et al. reviewed administration of small-molecule SHP-2 inhibitors for tumor therapies [134]. PTPs can also be involved in the progression of renal diseases. Amiri et al. found that Ang II could induce decreased SH2 domain-containing protein tyrosine phosphatase 1 (SHP-1) and increased SHP-2 under HG condition [16]. Li et al. recently found that protein tyrosine phosphatase non-receptor type 2 (PTPN2) could attenuate renal injury and fibrosis by suppressing STAT1/3-induced inflammation in early DN [135]. These studies revealed that clinical application of PTP agonists or inhibitors are promising in the treatment of renal diseases.

SOCS (SOCS1, SOCS2, SOCS3, SOCS4, SOCS5, SOCS6, SOCS7, and CIS) are induced to upregulate in response to ligands stimulation and inhibit ligands-induced STAT signaling pathways, forming a negative feedback loop. SOCS can inactivate the JAKs directly, block access of STATs to receptor binding sites, ubiquitinate and degrade signaling proteins and their subsequent targets [136]. SOCS1 binds to and inhibits JAKs directly. SOCS3 binds to JAK-proximal sites on cytokine receptors and inhibits JAK activity. CIS blocks the binding of STATs to cytokine receptors by competing with STATs for the same docking site [1,136]. SOCS can participate in renal diseases via modulating JAK/STAT pathways. SOCS3 was highly expressed in renal proximal tubules during AKI, which exacerbated AKI by inhibiting reparative renal epithelium regeneration and reparative macrophage phenotype transition via suppression of JAK/STAT pathways [137]. Anti-dsDNA IgG downregulated SOCS1 expression, activated JAK2/STAT1 signaling pathways and contributed to renal fibrosis in LN [138]. SOCS-1 and SOCS-3 could prevent IL-1β or OSM induced tubulointerstitial fibrosis in DN [85,86]. A cell-permeable peptide mimicking the kinase-inhibitory region of SOCS1 regulatory protein could halt the onset and progression of renal inflammation and fibrosis in DN by inhibiting STAT1 and STAT3 [139].

PIAS (PIAS1, PIAS3, PIASx, and PIASy) are constitutively expressed and interact with the dimeric form of STATs upon cytokine stimulation and inhibit STATs transcription activity through different methods: PIAS 1 and PIAS3 bind to and attenuate STAT1 and STAT3-mediated transcription activity by blocking DNA binding, respectively. PIASy and PIASx repress STAT1 and STAT4-mediated gene activation by recruiting other co-repressor molecules, such as histone deacetylases (HDACs), to inhibit transcription, respectively. The function of PIAS1 is controversial. PIAS1 might regulate STAT1-mediated IFN-responsive gene activation by promoting the SUMOylation of STAT1 on Lys703. Whether PIAS1 can regulate the STAT1 activity through SUMOylation depends on the promoter microenvironment of STAT1-target genes. PIAS1/y might repress transcription by sequestering transcription factors to certain subnuclear structures with enriched co-repressor complexes. This mechanism has not been confirmed in the regulation of STATs and needs to be further explored [140,141]. Yang et al. manifested upregulation of STAT3 with downregulation of PIAS3 in MRL/lpr mice [142]. In FSGS models, STAT1, STAT3, and PIAS1 mRNA levels were increased, and SOCS1, SOCS3, and PIAS3 mRNA levels were decreased at different time points [55]. It suggests that endogenous inhibitors may be involved in the progression of FSGS. What is more, downregulation of PIAS1 might participate in allograft rejection after kidney transplantation. PIAS1 might serve as a predictive marker for transplant fate [143].

## 6. Pharmacological Inhibition of STAT Signaling Pathways

Drug or ongoing drug clinical trials approved by the U.S. Food and Drug Administration (FDA) mainly target STAT signaling pathways and their upstream signaling elements such as Janus kinases and Src kinases. Although they have not yet been used in renal diseases, they may provide new strategies for renal disease treatment in the future. The specific pharmacological targets can be classified into cytokine and cytokine receptor inhibitors, kinase inhibitors, and STAT inhibitors. Drugs in clinical trials or approved by the FDA and their regulated renal diseases are shown in Table 3.

Inhibitors of IL-6R, such as tocilizumab and sarilumab, have been approved to treat rheumatoid arthritis and several other diseases. Other receptor inhibitors, such as VEGFR 2,3 and PDGFR inhibitors sorafenib, have also been investigated for the treatment of renal cell carcinoma [144]. The inhibitors of classical STAT upstream elements JAKs have been extensively studied; thus, there have been several drugs such as tofacitinib, ruxolitinib, baricitinib, fedratinib, upadacitinib, and AZD1480 in clinical trials or approved by the FDA undergoing clinical treatment. Src inhibitors, such as dasatinib and nintedanib, have been gradually studied and approved to treat leukemia and idiopathic pulmonary fibrosis, respectively. However, targeting upstream elements may bring serious side effects associated with inhibition of pleiotropic JAK/STAT signaling pathways. Furthermore, some kinase inhibitors have failed to show successful suppression of downstream STAT targets due to compensatory STATs activation mechanisms of other cytokines or upstream kinases [145,146]. Thus, targeting STAT molecules may be a more reasonable and viable therapy. Given the process of STATs functioning and the feasibility and effectiveness of drug targets, the specific targets of STATs mainly consist of SH2 domains and DNA-binding domains. Small-molecule inhibitors targeted the SH2 domains of STATs, such as STA-21, OPB-31121, TTI-101, and OPB-51602, have been tested, while specific inhibitors targeted DBD of STATs such as inS3–54 haven’t been investigated in the clinical trials due to their flat, “undruggable” structure, which can serve as a potential target for further development [147]. Furthermore, danvatirsen, a generation 2.5 antisense oligonucleotide (binds with STAT mRNA and silencse gene expression) under clinical development, has shown clinical activity in lymphoma [148]. Whether regulators directly targeted STAT domains or targeted endogenous inhibitors has not been tested in treatment of renal diseases yet. Most of the above inhibitors do not affect mitoSTAT3, making it an alternative pharmacological target. MDC-1112, phospho-valproic acid, was identified as a potential inhibitor of mitoSTAT3 by reducing its accumulation. Although MDC-1112 has been proven to be effective in pancreatic cancer and other solid tumors animal models, no clinical trials have been initiated yet [149]. Mitochondrial dysfunction induced by OPB-51602 (a SH2 domain-targeting STAT3 inhibitor) could lead to synthetic lethality in glucose-depleted cancer cells [150].

## 7. Conclusions

STAT signaling pathways are related to inflammatory reaction and immune response, and are thus involved in the pathological mechanisms of various renal diseases (Figure 3). As the name says, STAT signaling can transduce the signals of cytokines, chemokines, growth factors, and other different ligands from the cell membrane to the nucleus. As well, they can enable the cells to respond to external stimulation by activating cognate gene transcription or other non-canonical functions. Most studies have focused on the canonical STAT pathways and ignored the non-canonical pathways in renal diseases. In this review, the canonical and non-canonical STAT pathways are described, respectively. Non-canonical aspects of STAT signaling pathways in renal diseases are also illustrated. We also summarized the latest progression and clinical application of pharmacological inhibition of STAT signaling pathways. Although the current drug-targeted STAT-related molecules are mostly limited in the treatment of cancers and inflammation-driven diseases, clinical application of these drugs extending to renal diseases therapy in the future is promising.

## Figures and Tables

**Figure 1 cells-10-01610-f001:**
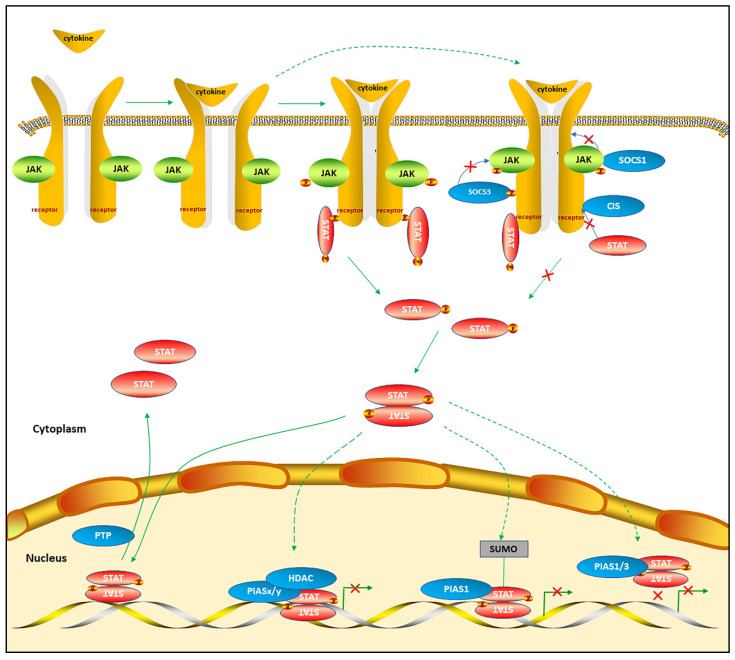
Canonical STAT signaling pathways (solid lines) and STAT endogenous inhibitors (dashed lines). Cytokines binding to cognate receptors causes the cytokine receptors dimerization. The caspase of Janus kinases (JAKs) and STATs are phosphorylated and activated. Phosphorylated STATs dissociate from the receptors, form homodimers or heterodimers, and transfer to the nucleus. Intranuclear STATs bind to specific DNA elements and activate the transcription of cytokine-responsive genes. Protein tyrosine phosphatases (PTPs) dephosphorylate activated pSTATs in the nucleus and dissociate STATs. Then, STATs are exported into the cytoplasm. Suppressors of cytokine signaling (SOCS) are upregulated in response to ligands stimulation and inhibit JAK/STAT pathways through different mechanisms: SOCS1 binds to and inhibits JAKs; SOCS3 binds to JAK-proximal sites on cytokine receptors and inhibits JAK activities; CIS competes and blocks STATs binding to the docking site of cytokine receptors. Constitutive protein inhibitors of activated STATs (PIAS) interact with pSTAT dimers in the nucleus upon cytokine stimulation and inhibit STAT transcription activity through different mechanisms: PIAS1/3 binds to STATs and blocks DNA binding; PIASy/x recruit other co-repressor molecules such as histone deacetylases (HDACs) to inhibit STATs-mediated gene transcription; PIAS1 may promote the SUMOylation of STATs.

**Figure 2 cells-10-01610-f002:**
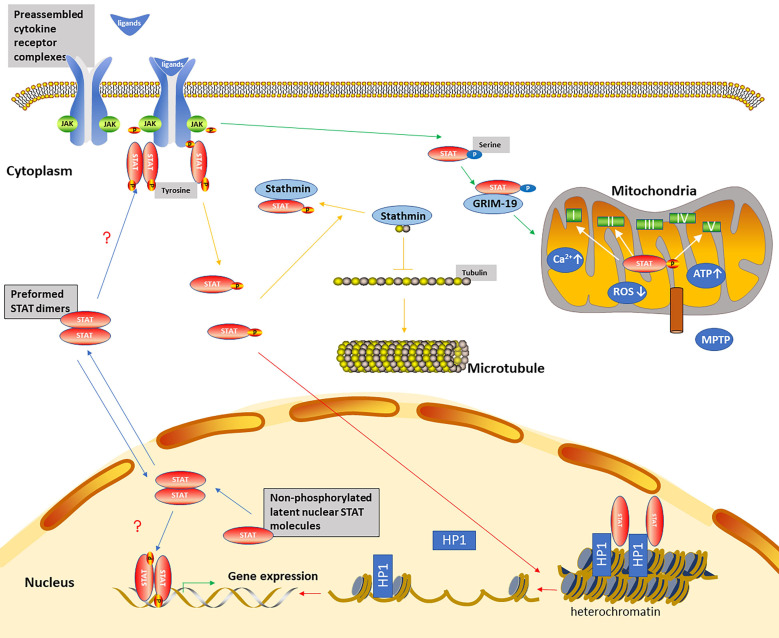
Non-canonical STAT signaling pathways. Non-canonical dynamics aspects (blue lines) include preassembled cytokine receptor complexes in the absence of ligands, preformed STAT dimers in the absence of the activating tyrosine, and non-phosphorylated latent nuclear STAT molecules consistently located in the nucleus. The question marks indicate that the mechanisms are unclear and still need to be further explored. Non-canonical functions include regulation of mitochondria (green lines), microtubule (yellow lines), and heterochromatin (red lines). Tyrosine phosphorylation of STATs promotes their serine phosphorylation. P-Ser-STAT3 is imported into mitochondria via GRIM-19. MitoStat3 promotes ATP synthesis, decreases ROS release, and increases mitochondrial Ca^2+^ and MPTP opening through regulating electron transport chains I, II, and V. Stathmin inhibits microtubule growth by binding tubulin dimers and sequestering tubulin, which decreases the concentration of free heterodimers available to polymerization. Phosphorylated STAT3 binding to stathmin inhibits its tubulin depolymerization activity. Unphosphorylated latent STAT binds to HP1 on heterochromatin in the nucleus. Phosphorylated STATs reduce the amount of unphosphorylated STAT localized on heterochromatin. HP1 disassociates from heterochromatin and leads to heterochromatin instability. Genes originally localized in heterochromatin are now accessible to STATs or other transcription factors.

**Figure 3 cells-10-01610-f003:**
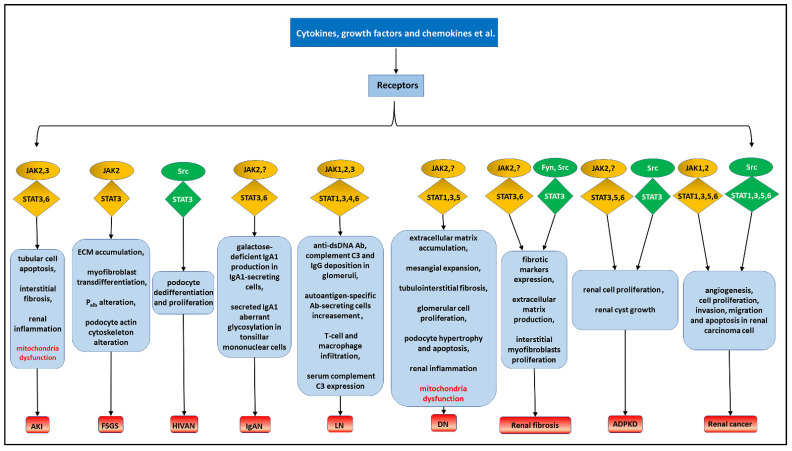
Overview of STAT signaling pathways involved in renal diseases. Ligands-receptors-kinases-STAT axis can mediate different biological events and regulate numerous renal diseases. Yellow circles: typical receptor tyrosine kinases JAKs. Green boxes: non-receptor tyrosine kinases SFK. Red fonts: non-canonical functions of STAT signaling pathways.

**Table 1 cells-10-01610-t001:** Main ligands and modulated biological events of STATs.

STATs	Main Ligands	Biological Events
STAT1	INFα/β, INFγ.	Anti-virus and anti-bacteria response; Cell growth; Cell apoptosis; Oncogenesis [1,8].
STAT2	INFα/β.	Anti-virus response; Oncogenesis [1,9].
STAT3	IL-6 family members: IL-6, IL-11, OSM, LIF, CLCF1, CNTF and erythropoietin, et al. Growth factors: EGF and HGF, et al.	Cell mitogenesis; Cell apoptosis; Oncogenesis; Cell proliferation; Th17 differentiation [1,7].
STAT4	IL-12.	Th1 development [1,10].
STAT5a	Prolactin, IL-2, GM-CSF, erythropoietin and other hormonelike cytokines.	Prolactin signaling; Treg cells differentiation [1,11].
STAT5b	Growth hormone, IL-2 and other hormonelike cytokines.	Growth hormone signaling [1,12].
STAT6	IL-4/13.	Th2 development [1].

INF, interferon; IL, interleukin; OSM, oncostatin M; LIF, leukemia inhibitory factor; CLCF, cardiotrophin-like cytokine factor; CNTF, ciliary neurotrophic factor; EGF, epidermal growth factor; HGF, hepatocyte growth factor; Th, T helper lymphocytes; GM-CSF, granulocyte-macrophage colony-stimulating factor; Treg, regulatory T lymphocyte.

**Table 2 cells-10-01610-t002:** Different roles of STAT pathways in Acute kidney injury.

Targets of STAT Pathways	AKI Exacerbation	AKI Amelioration
Ligands	IL-6, IL-19, CXCL8 [45,46,47,50]	IL-4/IL-13, IL-22 [48,49]
Receptor inhibitors		IL-6 inhibitors, CXCR1/2 antagonist: G31P, EGFR inhibitor: gefitinib [45,46,50,51]
JAK inhibitors	JAK3 inhibitor: AG490 [48]	JAK2 inhibitor: tofacitinib [44]
STAT inhibitors	STAT3 inhibitors [43]	STAT3 inhibitors [44]

AKI, acute kidney injury; IL, interleukin; CXCL, chemokine (C-X-C motif) ligand; CXCR, chemokine (C-X-C motif) receptor; EGFR, epidermal growth factor.

**Table 3 cells-10-01610-t003:** Pharmacological Inhibition of STATs in renal diseases.

Targeted Sites	Drug in Clinical Trials or Approved by FDA	Related Renal Diseases
Cytokine and cytokine receptor inhibitors	IL-6R inhibitors: tocilizumab, sarilumab	Tocilizumab in kidney transplantation [151]
	VEGFR& EGFR inhibitors: Sorafenib	Sorafenib in renal cell carcinoma [144]
Kinase inhibitors	JAK inhibitors: tofacitinib, ruxolitinib, baricitinib, fedratinib, upadacitinib (ClinicalTrials.gov) AZD1480 (NCT00910728)	Tofacitinib in kidney transplantation [152] Tofacitinib in end-stage renal disease undergoing hemodialysis [153] Baricitinib in type 2 diabetes and diabetic nephropathy [154]
	Src inhibitors: dasatinib [145], Nintedanib [146]	Dasatinib in senescence of Chronic Kidney Disease (NCT02848131) Nintedanib in renal cell carcinoma [155]
STAT inhibitors	Inhibitors targeted SH2 domain: STA-21(NCT01047943), OPB-31121 (NCT01406574), TTI-101(NCT03195699), OPB-51602 (NCT02058017)	
	Antisense oligonucleotide: danvatirsen [148]	

FDA, the U.S. Food and Drug Administration; VEGFR, vascular endothelial growth factor receptor; EGFR, epidermal growth factor receptor; JAK, janus kinases; DBD, DNA binding domain. Drug’s data are from Drugs@FDA: FDA-Approved Drugs or ClinicalTrials.gov.

## Data Availability

All data is included in the manuscript.

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
