# Peer review of "Targeting Canonical and Non-Canonical STAT Signaling Pathways in Renal Diseases"

_cells, 2021, doi:10.3390/cells10071610_

Round 1

Reviewer 1 Report

the authors of the review on "Targeting canonical and non-canonical STAT signaling pathways in kidney disease" describe the topic comprehensively. A note, from the data in the literature the canonical pathways of STATs are not only those mediated by JAKs receptors but also include all those mediated by other receptors with tyrosine kinase activity (such as EGFR, PDGFR, etc.) and those mediated by cytoplasm kinase and GPCR. Instead, the non-canonical pathways of STATs indicate all other cellular activities of STAT proteins such as mitochondrial ones, stabilization of the cytoskeleton, etc. In my opinion, these paragraphs need to be revisited, as they concern the core topic of the review.

Author Response

The authors of the review on "Targeting canonical and non-canonical STAT signaling pathways in kidney disease" describe the topic comprehensively. A note, from the data in the literature the canonical pathways of STATs are not only those mediated by JAKs receptors but also include all those mediated by other receptors with tyrosine kinase activity (such as EGFR, PDGFR, etc.) and those mediated by cytoplasm kinase and GPCR. Instead, the non-canonical pathways of STATs indicate all other cellular activities of STAT proteins such as mitochondrial ones, stabilization of the cytoskeleton, etc. In my opinion, these paragraphs need to be revisited, as they concern the core topic of the review.

Reply:We thank the reviewer for giving us this valuable suggestion. We redefined the scope of canonical (line 59-117, page 2-3) and non-canonical STAT pathways (line 132-185, page 4-5 ) and discussed about them in the revised manuscript (line 225-227, page 7, reference 52; line 341-346, page 9, reference 95; line 546-552, page 13, reference 149-150).

Reviewer 2 Report

With great interest I read the review “Targeting Canonical and Non-canonical STAT signalling pathways in renal diseases” by Lili Gai et al.

The authors analyse the impact canonical and non-canonical STAT pathways in renal diseases and how they may be modulated by endogenous inhibitors. Moreover, the review describes the clinical implications of pharmacological inhibitors tested in clinical trials or approved by FDA for the treatment of renal diseases.

The review is detailed, very informative and supported by useful figures and tables. However, extensive editing of English language and style are required for some paragraphs.

In addition, I would include a paragraph describing the role of STAT signalling pathway in renal tumors and potential clinical implication of STAT inhibitors.

Author Response

The review is detailed, very informative and supported by useful figures and tables. However, extensive editing of English language and style are required for some paragraphs. In addition, I would include a paragraph describing the role of STAT signaling pathway in renal tumors and potential clinical implication of STAT inhibitors.

Reply:We thank the reviewer for giving us the suggestions. We have gone through the whole paper carefully and tried our best to edit the English language. And accordingly, we added a paragraph describing the role of STAT signaling pathway in renal tumors and potential clinical implication of STAT inhibitors and they are presented in line 402-431, page 10-11.

Reviewer 3 Report

I have the following suggestions.

  • Are cancers and Autoimmune related diseases?  (line 36)
  • Please add a reference to table 1.
  • The manuscript lacks reference in several places.
  • A table about which of the STAT inhibitor treatment increases the AKI and which one ameliorates it will be good for clear visualization.
  • It will be good to differentiate canonical and noncanonical STAT signaling pathways with different colors in Figure 3.

Author Response

  • Are cancers and Autoimmune related diseases?  (line 36)

Reply: We thank the reviewer for this valuable suggestion. We added references about STAT signaling pathways involved in cancers (reference 5). We distinguished cancers from autoimmune diseases and listed them as follows: ‘they are involved in many inflammatory and immune related diseases including cancers and some autoimmune diseases such as systemic lupus erythematosus, inflammatory bowel diseases and rheumatological diseases (line 38-41).’

  • Please add a reference to table 1.

Reply: We have searched and added related references in table 1.

  • The manuscript lacks reference in several places.

Reply: Sorry for this negligence. We have gone through the whole paper carefully and added references after some places in the manuscript.

  • A table about which of the STAT inhibitor treatment increases the AKI and which one ameliorates it will be good for clear visualization.

Reply: We thank the reviewer for this valuable suggestion. Accordingly, we added a table as Table 2. to summary the different roles of STAT inhibitor treatments in AKI.

  • It will be good to differentiate canonical and noncanonical STAT signaling pathways with different colors in Figure 3.

Reply: Thanks for the review’s advice. We adjusted the figures to differentiate canonical and noncanonical STAT signaling pathways.

Round 2

Reviewer 1 Report

The current version of the review in my opinion can be published.